# Modified ADRC Design of Permanent Magnet Synchronous Motor Based on Improved Memetic Algorithm

**DOI:** 10.3390/s23073621

**Published:** 2023-03-30

**Authors:** Gang Liu, Chuanfang Xu, Longda Wang

**Affiliations:** 1College of Engineering, Inner Mongolia Minzu University, Tongliao 028000, China; 2School of Mechanical and Electrical Engineering, Jiangxi New Energy Technology Institute, Xinyu 338004, China; 3Department of Automation, Shanghai Jiao Tong University, Shanghai 200240, China; 4School of Automation and Electrical Engineering, Dalian Jiaotong University, Dalian 116026, China

**Keywords:** permanent magnet synchronous motor, auto disturbance rejection control, improved memetic algorithm, optimal control function, Gaussian mutation, fusion distance

## Abstract

In this paper, a novel modified auto disturbance rejection control (ADRC) design of a permanent magnet synchronous motor based on the improved memetic algorithm (IMA) is proposed. Firstly, there is an obvious system ripple caused by the defect that the optimal control function used in traditional ADRC cannot be differentiable and smooth at the segment point; aiming at weakening the system ripple effectively, the proposed method constructs a novel differentiable and smooth optimal control function to modify the ADRC design. Furthermore, aiming at improving the integration parameters optimization effect effectively, a novel improved memetic algorithm is proposed for obtaining the optimal parameters of ADRC. Specifically, an IMA with high-quality balance based on an adaptive nonlinear decreasing strategy for the convergence factor, Gaussian mutation mechanism, improved learning mechanism with the high-quality balance between competitive and opposition-based learning (OBL) and an elite set maintenance mechanism based on fusion distance is proposed so that these strategies can improve the optimization precision by a large margin. Finally, the experiment results of the PMSM speed control practical cases show that the ADRC based on IMA has an apparent better optimization effect than that of fuzzy PI, traditional ADRC based on the genetic algorithm and an improved ADRC based on improved moth–flame optimization.

## 1. Introduction

The permanent magnet synchronous motor is widely employed in industry fields of new energy electric vehicles, CNC machine tools, subway vehicles, etc., due to having such advantages as high power density, easy maintenance, simple structure, and convenient speed adjustment [1]. PI and ADRC are two popular control algorithms for PMSM. PI control has the advantages of simple structure and easy implementation, and ADRC has the advantages of high precision and strong stability [2,3]. However, the tracking control accuracy is relatively low by using PI and traditional ADRC in PMSM speed control. The purpose of this paper is to enhance the control performance for active disturbance resisting controller (ADRC) for PMSM so as to obtain a more ideal PMSM speed control quality.

For improving the performances of active disturbance rejection controller (ADRC) for PMSM, scholars have proposed some improvement strategies. An ADRC strategy of the signal injection-based interior PMSM drive was raised in [4]. A class of linear–nonlinear switching ADRCs to design speed controllers and current controllers for PMSM in servo systems was investigated in [5]. An ADRC solution of the angular velocity trajectory tracking task for basic disturbance, uncertainty and PMSM is presented in [6]. An enhanced active disturbance rejection control (ADRC) combined with quasi-resonant controllers (QRCs) for the PMSM speed loop was proposed in [7]. A discrete-time repetitive control-based active disturbance rejection control (ADRC) for the current loop of PMSM drives was proposed in [8]. An active disturbance rejection controller (ADRC) based on deep reinforcement learning (DRL) algorithm was proposed to be used in the flux weakening control (FWC) system of motors for more electric aircraft in [9]. A class of linear–nonlinear switching active disturbance rejection control (ADRC) to design speed controllers and current controllers for permanent magnet synchronous machine (PMSM) in servo systems was investigated in [10]. The active disturbance rejection control (ADRC) and feedback compensation control method that can solve the speed fluctuation problem of permanent magnet synchronous motors was proposed in [11]. A novel mirror milling trajectory planning method based on fuzzy-ADRC controlled force pre-supporting was proposed in [12]. A novel Nonlinear Consequent Part Recurrent Type-2 Fuzzy System (NCPRT2FS) was presented for the modeling of renewable energy systems in [13]. A novel mirror milling trajectory planning method based on fuzzy-ADRC controlled force pre-supporting was proposed in [12]. A novel Nonlinear Consequent Part Recurrent Type-2 Fuzzy System (NCPRT2FS) is presented for the modeling of renewable energy systems in [13]. For systems with uncertainties, time-varying delays, unknown disturbances, as well as strong nonlinearity, a robust fuzzy predictive control (RFPC) based on the Takagi–Sugeno (T-S) fuzzy model was proposed in [14]. A robust adaptive model predictive control (RMPC) with an underlying discrete-time adaptive controller was proposed in [15]. Obviously, the existing studies can improve the control effect of ADRC for PMSM.

In addition, the key parameters of ADRC are extremely important for the effect of PMSM velocity tracking control. However, there are many complex uncertain factors and relationships between them, so it is easy to fall into a local convergence if only using some traditional optimization algorithms, such as the moth–flame optimization algorithm, genetic algorithm, memetic algorithm, whale optimization algorithm, etc. To solve the aforementioned issue in automatic traditional optimization algorithms, many works in the literature discussed the traditional optimization algorithms. An efficient moth–flame optimization algorithm was proposed to solve the distributed generations and shunt capacitor banks optimization problems in [16]. A hybrid learning algorithm that combines the genetic algorithm (GA) with gradient descent (GD) was proposed in [17]. A novel memetic algorithm using modified particle swarm optimization (PSO) for PMSM design was proposed in [18]. An effective memetic algorithm for curvature-constrained path planning of messenger UAV in air–ground coordination was proposed in [19]. A memetic evolutionary multi-objective optimization method for the power unit commitment problem was proposed in [20]. A whale optimization algorithm to tackle the three-dimensional path planning of autonomous underwater vehicles was proposed in [21]. An improved whale optimization algorithm based on the Tchebycheff decomposition method, convergence factor nonlinear decline strategy, and genetic evolution measurement for model predictive controller was proposed in [22]. However, there are few related works published on the active disturbance rejection controller based on the effective memetic algorithm for PMSM.

As for the problem that the existing active disturbance rejection controllers of PMSM controllers do not have sufficient competence, this paper proposes a novel modified ADRC of PMSM based on improved memetic algorithm (IMA). The following summarizes the main contributions of this paper:

(I) An improved memetic algorithm (IMA) based on an adaptive nonlinear decreasing convergence factor strategy for the whale optimization algorithm, a Gaussian mutation for simulated annealing, a learning mechanism using mixtures of competitive mechanism and OBL mechanism and an elite set maintenance mechanism based on fusion distance are proposed.

(II) A novel differentiable and smooth nonlinear function is constructed for modifying ADRC for overcoming the non-differentiable and unsmoothed situation at the piecewise point of nonlinear functions in the conventional ADRC; the above IMA is proposed for improving the effectiveness for the optimization integration of ADRC key parameters so as to improve the performance for modified ADRC of PMSM based on IMA effectively.

For the proposed novel modified ADRC of PMSM based on the improved memetic algorithm (IMA), the following summarizes the major novelties of this paper:

(I) An improved memetic algorithm (IMA) with the high-quality global optimization performance is proposed. In this proposed algorithm, an improved whale optimization algorithm (IWOA) by adopting an adaptive nonlinear decreasing strategy for convergence factor is designed for global search, an improved simulated annealing (ISA) by introducing the Gaussian mutation mechanism is designed for local development, and an improved learning mechanism between the elite set and common individual population based on competitive and opposition-based learning and an elite set maintenance mechanism based on fusion distance and congestion degree distance are incorporated for improving global optimization performance effectively.

(II) Based on the traditional optimal control function, inverse hyperbolic sine function and sine function are introduced, and a novel optimal control function is constructed by the fitting method for modifying ADRC.

This research is structured as follows: Section 2 presents an introduction to the ADRC for PMSM. Section 3 illustrates several intelligent optimization algorithms, including the memetic algorithm, whale optimization algorithm and simulated annealing. Section 4 introduces the modified ADRC redesigned and IMA proposed in this paper. Section 5 presents the experimental outcomes and performs the corresponding analysis. Section 6 sums up this paper.

## 2. Active Disturbance Rejection Controller for Permanent Magnet Synchronous Motor

### 2.1. Permanent Magnet Synchronous Motor Model

The surface mounted structure is widely used in PMSM because the approximate sine wave distribution for air gap flux density waveform about the motor can be obtained, and the inductance components of the direct axis (*d* axis) and the quadrature axis (*q* axis) of the motor can be equal, thus ensuring that the motor has a good operation performance. The specific voltage equation of surface mounted structure PMSM in two-phase rotating coordinate system is as follows:(1)$$\begin{array}{c} \left\{\begin{array}{c} u_{sd}=R_{s}i_{sd}+\frac{d\psi_{sd}}{dt}-\omega_{r}\psi_{sq}\\ u_{sq}=R_{s}i_{sq}+\frac{d\psi_{sq}}{dt}+\omega_{r}\psi_{sd} \end{array}\right. \end{array}$$
where $u_{sd}$, $u_{sq}$, $i_{sd}$, $i_{sq}$, $\psi_{sd}$, $\psi_{sq}$ refer to the direct and quadrature axial components in motor stator voltages, stator currents, stator flux under two-phase rotation coordinate system dq, and $\omega_{r}$ is the rotation angle of the rotor flux.

The specific calculation formula of stator flux is as follows:(2)$$\begin{array}{c} \left\{\begin{array}{c} \psi_{sd}=L_{d}i_{sd}+\psi_{f}\\ \psi_{sq}=L_{q}i_{sq} \end{array}\right.\; \end{array}$$
where $L_{d}$ and $L_{q}$ are the inductance, and $\psi_{f}$ is the flux linkage of the permanent magnet.

The motor torque equation is as follows:(3)$$\begin{array}{c} T_{e}=\frac{3}{2}p\left(\psi_{sd}i_{sq}-\psi_{sq}i_{sd}\right)=\frac{3}{2}p\left[\psi_{f}i_{sq}+\left(L_{d}-L_{q}\right)i_{sq}i_{sd}\right] \end{array}$$
where $T_{e}$ represents the electromagnetic torque.

The mechanical motion equation is as follows:(4)$$\begin{array}{c} T_{e}-T_{L}=J\frac{d\omega_{m}}{dt} \end{array}$$
where $T_{L}$ represents the load torque, *J* represents the moment of inertia, and $\omega_{m}$ represents the motor velocity.

### 2.2. Design of the Active Disturbance Rejection Controller

The active disturbance rejection controller (ADRC) mainly consists of the tracking differentiator (TD), extended state observer (ESO) and nonlinear state error feedback (NLSEF).

The specific calculation formula of TD is as follows:(5)$$\begin{array}{c} TD\left\{\begin{array}{c} v_{1}\left(k+1\right)=v_{1}\left(k\right)+hv_{2}\left(k\right)\\ v_{2}\left(k+1\right)=v_{2}\left(k\right)+hfst(v_{1}\left(k\right)-v_{0},v_{2}\left(k\right),r,h_{0} \end{array}\right. \end{array}$$
where hfst represents the TD function set; $v_{1}$ represents the tracking value of expected speed; $v_{2}$ represents the differential of $v_{1}$; *k* represents the number of current periods; *h* represents the step length; $h_{0}$ represents the filter coefficient; *r* represents the speed factor, the value affects the speed of TD process; and $v_{0}$ represents the initial state of $v_{1}$ [23].

ESO is the core of the ADRC, and the specific calculation formula of ESO is as follows:(6)$$\begin{array}{c} ESO\left\{\begin{array}{c} e_{1}=z_{1}\left(k\right)-y\\ z_{1}\left(k+1\right)=z_{1}\left(k\right)+h\left(z_{2}\left(k\right)-\beta_{01}e_{1}\right)\\ z_{2}\left(k+1\right)=z_{2}\left(k\right)+h\left(z_{3}\left(k\right)-\;\beta_{02}fal\left(e_{1},\alpha_{1},\delta_{1}\right)+b_{0}u\right)\\ z_{3}\left(k+1\right)=z_{3}\left(k\right)-h\beta_{03}fal\left(e_{2},\alpha_{2},\delta_{2}\right) \end{array}\right. \end{array}$$
where fal represents the optimal control function; $\beta_{01}$, $\beta_{02}$, $\beta_{03}$, $\alpha_{1}$, $\alpha_{2}$, $b_{0}$, $\delta_{1}$, $\delta_{2}$ are parameters that need to be adjusted in ESO; *y* represents the output; $z_{1}$, $z_{2}$ and $z_{3}$ represent the state variable; $e_{1}$ and $e_{2}$ represent the error between the TD result and state variable [24].

The specific calculation formula of the traditional optimal control function fal(e,α,δ) is as follows: (7)fal(e,α,δ)=eαsignee>δeeδ1−αδ(1−α)e≤δ

.

The differential form of the derivative function for optimal control function fal(e,α,δ) is as follows:(8)$$\begin{array}{c} fal^{\prime }\left(e,\alpha ,\delta \right)=\left\{\begin{array}{c} \frac{1}{\delta^{1-\alpha }}\;\;\;\;\;\;\;\;\;\;\;\;\;\;\;\;\;\;\;,\;\;\;0<e\leq \delta \\ \alpha e^{\alpha -1}\;\;\;\;\;\;\;\;\;\;,\;\;\;\;\delta <e \end{array}\right. \end{array}$$

At the segment point δ, the value of the function $fal^{\prime }\left(e,\alpha ,\delta \right)$ is
(9)$$\begin{array}{c} fal^{\prime }\left(\delta_{-},\alpha ,\delta \right)=\frac{1}{\delta^{1-\alpha }} \end{array}$$
(10)$$\begin{array}{c} fal^{\prime }\left(\delta_{+},\alpha ,\delta \right)=\alpha \delta^{\alpha -1} \end{array}$$

When the function fal(e,α,δ) is differentiable in the segment, restraint condition $fal^{\prime }\left(\delta_{-},\alpha ,\delta \right)=fal^{\prime }\left(\delta_{+},\alpha ,\delta \right)$ should be met. In this case, parameter α=1, and function fal(e,α,δ)=1, lead to destroying the nonlinear characteristics of the optimal control function. Thus, redesigning the optimal control function is clearly necessary.

The specific calculation formula of NLSEF is as follows:(11)$$\begin{array}{c} NLSEF\left\{\begin{array}{c} e_{1}\left(k\right)=v_{1}\left(k\right)-z_{1}\left(k\right)\\ e_{2}\left(k\right)=v_{2}\left(k\right)-z_{2}\left(k\right)\\ u_{0}=\beta_{21}fal\left[e_{21}\left(t\right),\alpha_{21},\delta_{21}\right]\\ \;\;\;\;\;\;\;\;\;\;\;\;\;\;\;+\beta_{22}fal\left[e_{22}\left(t\right),\alpha_{22},\delta_{22}\right]\\ u=u_{0}-\frac{z_{3}}{b_{0}} \end{array}\right. \end{array}$$
where $e_{21}$ is the error between the follow value of the TD to the reference signal and the estimated value of the system output signal by the ESO; $e_{22}$ is the differential of $e_{21}$; *u* represents the output of NLSEF; $u_{0}$ represents the control rate; and $b_{0}$ is the compensation coefficient to eliminate the interference signal [25].

The control structure diagram about the above active disturbance rejection controller is shown in Figure 1.

The above narrative shows that the adjusted parameters ($\beta_{01}$, $\beta_{02}$, $\beta_{03}$, $\beta_{1}$, and $\beta_{2}$) are significant for ADRC. In order to obtain the appropriate adjusted parameters efficiently, the ADRC parameters adjusting method using the intelligent optimization algorithm is proposed.

### 2.3. Design of PMSM Speed Control System Based on ADRC

The PMSM speed control system based on ADRC is composed of an ADRC, an inverter circuit and a PMSM. The ADRC controls the speed and torque of the PMSM. The specific meaning of the input and output signals of each module in the PMSM speed control system needs to be determined according to the controlled object. The transformation equation about the PMSM of Equations (2)–(4) is as follows:(12)$$\begin{array}{c} \frac{d^{2}\omega_{m}}{dt^{2}}=\frac{1}{J}\left[\begin{array}{c} -\frac{dT}{dt}-B\frac{d\omega_{m}}{dt}\\ +\frac{3p_{n}\psi_{f}}{2L_{s}}\left(-R_{s}i_{q}-p_{n}\psi_{f}\omega_{m}+u_{q}\right) \end{array}\right] \end{array}$$

The greatest advantage of ADRC is its capability in estimating disturbances inside and outside of the system, and making a precise compensation. For obtaining the information of control quantity and disturbance quantity, the further transformation equation about PMSM is as follows:(13)$$\begin{array}{c} \left\{\begin{array}{c} \frac{d^{2}\omega_{m}}{dt^{2}}=\frac{3p_{n}\psi_{f}}{2JL_{s}}u_{q}+\alpha \left(t\right)\\ \alpha \left(t\right)=\frac{1}{J}\left[\begin{array}{c} -\frac{dT}{dt}-B\frac{d\omega_{m}}{dt}\\ -\frac{3p_{n}\psi_{f}}{2L_{s}}\left(R_{s}i_{q}+p_{n}\psi_{f}\omega_{m}\right) \end{array}\right]+f\left(t\right) \end{array}\right. \end{array}$$
where f(t) represents the unobservable disturbance of the system.

According to the PMSM mathematical model, ADRC designed principle and vector control frame, a PMSM speed control system based on an improved ADRC is acquired. The control block diagram of PMSM speed control system based on ADRC is shown in Figure 2.

According to Figure 3, the ADRC variable related to the control system quantity is realized, and the PMSM speed control system will have better performance in robustness. The specific forms of variables in TD, ESO and NLSEF expressions are gained.

As for TD, the real meaning of the reference signal *v* in controller refers to the expected velocity $\omega_{m}^{*}$. The real meaning of output signal $x_{1}$ within control system symbolizes the tracking signal $\omega_{ref}$ related to expected velocity based on TD. As for ESO, the realistic importance for the output signal *y* in the control system means real speed $\omega_{m}$; the practical significance of the output signal $z_{2}$ in the control system is the actual speed tracking signal; and the practical significance of the output signal $z_{3}$ in the control system is the estimation of the disturbance signal. As for NLSEF, realistic significance in input signal within control system can be evidenced by the error of $\omega_{ref}$, ${\hat{\omega }}_{ref}$, ${\dot{\omega }}_{ref}$ and ${\hat{\dot{\omega }}}_{ref}$.

Generally, $\beta_{11}=3\omega_{0}$, $\beta_{12}=3\omega_{0}^{2}$, $\beta_{13}=3\omega_{0}^{3}$, $\beta_{12}=3\omega_{c}^{2}$, $\beta_{22}=2\zeta \omega_{c}$ for the second-order ADRC of PMSM, and $\omega_{0}$, $\omega_{c}$ refer to observer and controller bandwidth, separately [26]. In practice, 6 key parameters (*r*, *h*, $\Delta_{11}$, $\Delta_{12}$, $\Delta_{21}$, $\Delta_{22}$ and $b_{0}$) are usually set by an empiric value, and six other key parameters ($\alpha_{11}$, $\alpha_{12}$, $\alpha_{21}$, $\alpha_{22}$, $\omega_{0}$ and $\omega_{c}$) are necessary for intelligent optimization integration.

### 2.4. Evaluation Model of PMSM Speed Control

The integral of time multiplied by the absolute value of error (ITAE) is one of the significant evaluation indexes for ADRC control performance. The specific calculation formula of ITAE is as follows:(14)$$\begin{array}{c} ITAE=\int t\left|e\left(t\right)\right|dt. \end{array}$$

During the PMSM speed tracking control process, the time about the long-term control range with the acceptable maximum absolute value of speed and torque error ($\left|SE_{S}max\right|$ and $\left|SE_{T}max\right|$) is called stable time St, and the above referred error should be less than maximum acceptable stable control speed and torque error ($Amax(SE_{S})$ and $Amax(SE_{T})$); the time between the beginning time point of power-up and the corresponding begin time point of the stable control range is called the adjusting time At (adjusting time At must not exceed maximum acceptable adjusting time Amax(At)); the maximum absolute value of the speed and torque error ($\left|E_{S}max\right|$ and $\left|E_{T}max\right|$) should be less than the acceptable maximum absolute value of the speed and torque error ($Amax(E_{S})$ and $Amax(E_{T})$). In addition, the speed and torque ITAE ($ITAE_{S}$ and $ITAE_{T}$) for the whole PMSM speed tracking control process should be less than the maximum acceptable speed and torque ITAE ($Amax(ITAE_{S})$ and $Amax(ITAE_{T})$). The specific evaluation model of PMSM speed control is as follows:(15)$$\begin{array}{c} \left\{\begin{array}{c} \min\left(ITAE_{S},ITAE_{T},\left|SE_{S\max}\right|,\left|SE_{T\max}\right|,\left|E_{S\max}\right|,\left|E_{T\max}\right|\right)\\ \;\;\;\;\left|SE_{S\max}\right|<Amax\left(SE_{S}\right),\left|SE_{T\max}\right|<Amax\left(SE_{T}\right)\\ s.t.\;\left|E_{S\max}\right|<Amax\left(E_{S}\right),\left|E_{T\max}\right|<Amax\left(E_{T}\right)\\ \;\;\;\;ITAE_{S}<Amax\left(ITAE_{S}\right)\\ \;\;\;\;ITAE_{T}<Amax\left(ITAE_{T}\right)\\ \;\;\;\;At<Amax\left(At\right) \end{array}\right. \end{array}$$
where $ITAE_{S}$, $ITAE_{T}$, $\left|SE_{S\max}\right|$, $\left|SE_{T\max}\right|$, $\left|E_{S\max}\right|$, $\left|E_{T\max}\right|$ are six performance evaluation indexes for optimization.

Based on these above performance evaluation indexes and boundary constraints, the diagram about the above parameters adjusting method using intelligent optimization algorithm (IOA) is shown in Figure 3.

## 3. Intelligent Optimization Algorithms

### 3.1. Memetic Algorithm

As early as 1976, meme was proposed by R. Dawkins. In the theory of meme, meme is the basic unit of culture, and it is spread through imitation and learning, which is passed on from generation to generation. The memetic algorithm was proposed by Pablo Moscato in 1989 on the basis of the theory of meme [27]. In fact, the memetic algorithm proposes a framework, which can be equal to a collaboration model between the global population evolution and local individual learning. The model of the memetic algorithm is similar to the genetic algorithm; however, under the premise of an appropriate framework design, its global optimization performance far exceeds that of the genetic algorithm, and for several specific optimization problems, the optimization precision could be improved to a considerable extent, even by several orders of magnitudes [28]. For the motivation of improving the integration parameters optimization effect effectively, the memetic algorithm is used in this paper, and its improvement strategies study is also heeded.

### 3.2. Whale Optimization Algorithm

WOA simulating humpback whale foraging refers to a novel heuristic optimization algorithm [29]. It is composed of three parts: surrounding prey, bubble hunting, and searching for prey. In ocean activities, hump-back whales have a special hunting strategy such that they make distinctive bubbles along the circular path or the path with the shape of nine to keep them close to their prey.

The mathematical model of surrounding prey and bubble hunting is as follows:(16)$$\begin{array}{c} X\left(t+1\right)=\left\{\begin{array}{c} X^{*}\left(t\right)-A\cdot D\;\;\;\;\;\;\;\;\;\;\;\;\;\;\;\;\;\;\;\;\;\;\;p\leq p_{s}\\ X^{*}\left(t\right)+D_{p}\cdot e^{bl}\cdot \cos\left(2\pi l\right)\;\;\;\;\;\;\;\;p>p_{s} \end{array}\right. \end{array}$$
where *p* represents the probability for behavior selection of humpback whales, p∈[0,1]; $p_{s}$ represents the probability for surrounding prey, $p_{s}\in \left[0,1\right]$ and $1-p_{s}$ is the probability for bubble hunting; $D_{p}=\left|CX^{*}\left(t\right)-X\left(t\right)\right|$ represents the absolute value of difference between $CX^{*}\left(t\right)$ and x(t); $X^{*}\left(t\right)$ represents the best position vector at present; x(t) represents the whale position; *b* is the constant relating to the logarithmic spiral morphology, the value is 1; *l* is the random number in (−1,1); *A* and *C* are the coefficient; $A=2a\times r_{1}-a$, $C=2\times r_{2}$, $a=2-2\times t/T_{\max}$, $r_{1}$ and $r_{2}$ are random numbers in (−1,1); *t* represents the number of current iteration times; and $T_{\max}$ represents the maximum number of iteration times.

The mathematical model of searching for prey is as follows:(17)$$\begin{array}{c} \left\{\begin{array}{c} D=\left|CX_{rand}-X\left(t\right)\right|\\ X\left(t+1\right)=X_{rand}-A\cdot D \end{array}\right. \end{array}$$
where $D=\left|CX_{rand}-X\left(t\right)\right|$ represents the absolute value of difference between $CX_{rand}$ and x(t); $X_{rand}$ represents the randomly selected position vector of the whale, $\left|X_{rand}-X\left(t\right)\right|\leq \Delta$; and Δ represents the neighbor-hood areas of X(t) [30].

### 3.3. Simulated Annealing

Aiming at solving the practical combinatorial optimization problem effectively, Kirk-Patrick et al. adopted simulated annealing firstly. According to the cooling process of metal, simulated annealing is used to generate a large number of superior solutions with the continuous decrease in temperature, and parts of them are accepted [31]. The specific evolution circumstances should be subordinated to the Metrolips standard. In addition, the current temperature of the metal is decreased gradually during the whole iteration; if it is below the threshold temperature, the calculation of simulated annealing will be over. So, the regulation design significance of Metrolips and temperature reduction are necessary to be honored.

The regulation design of Metrolips adopted in this paper is as follows:(18)$$\begin{array}{c} \left\{\begin{array}{c} MP=\exp\left(\frac{fit\left(X(u)\right)-fit\left(X_{e}\left(t,q\right)\right)}{T_{c}\left(k\right)}\right)\\ MP>1-rand(0,MT) \end{array}\right. \end{array}$$
where $fit(X_{e}\left(t,q\right))$ represents the fitness function value of aboriginal solution $X_{e}\left(t,q\right)$ before the search process of simulated annealing, and it is the q-th elite individual, fit(X(t)) represents the fitness function value of generated solution X(u) in the neighborhood of aboriginal solution X(t), exp(X) represents an exponential function based on natural constant e≈2.718, $T_{c}$ represents the current temperature of the k-th period, MP represents the annealing probability, MT represents the appropriate threshold value for Metrolips standard, and rand(0,MT) represents a random number in (0,MT).

The regulation design of temperature reduction in this paper is as follows:(19)$$\begin{array}{c} T_{c}\left(k+1\right)=T_{c}\left(k\right)\times \alpha_{T}\times min\left(1,MP\right); \end{array}$$
where $\alpha_{T}$ represents the cooling coefficient.

At the initialization time, initial temperature $T_{0}$ should be assigned, and the current temperature $T_{c}\left(0\right)$ is equal to $T_{0}$; the above temperature reduction and Metrolips regulations should be obeyed. If the current temperature $T_{c}$ is below the threshold temperature $T_{end}$, the iteration will be over.

## 4. ADRC Based on Improved Memetic Algorithm

### 4.1. Design of a Novel Differentiable and Smooth Nonlinear Optimal Control Function

The optimal control function for ADRC should be differentiable at the piecewise point and smooth to the uttermost extent. The actual value for Δ is always tiny in engineering. If the optimal control function is not differentiable or not sufficiently smooth at the piecewise point, when Δ is tiny, there are great system amplitude output oscillations, which are detrimental to the improved system performance quality. A new novel differentiable nonlinear optimal control nfal(e,α,δ) based on primitive function fal(e,α,δ) by the function of inverse hyperbolic arsinhe and tangent tane is proposed in [32]. However, as far as nfal(e,α,δ), the tangent function is not enough smooth, key parameter α is not involved, and it will not be conducive to the effective improvement of ADRC.

Based on the above analysis, this paper constructs a novel differentiable and smooth optimal control function newfal(e,α,δ) based on primitive function fal(e,α,δ) by the function of inverse hyperbolic arsinhe and sine sinαe. The specific details of the design can be seen as given below.

As for the issues about the non-smooth linear segment and the non-differentiable at the piecewise point in function fal(e,α,δ) in the case of |e|≤δ, it is replaced by the linear function containing the inverse hyperbolic sin*e* function with superior smoothness and sin αe function in this paper.

The specific formula about the novel optimal control function newfal(e,α,δ) with |e|>δ is as follows:(20)$$\begin{array}{c} newfal\left(e,\alpha ,\delta \right)=\mathrm{sign}\left(e\right)|e{|}^{\alpha } \end{array}$$

The specific formula about the novel optimal control function newfal(e,α,δ) with |e|≤δ is as follows:(21)$$\begin{array}{c} newfal\left(e,\alpha ,\delta \right)=a_{1}\cdot \mathrm{arsinh}e+a_{2}\cdot \sin\alpha e \end{array}$$

If |e|≤δ, the optimal control function newfal(e,α,δ) comprises the inverse hyperbolic sine function and sine function, and it is capable of guaranteeing the constantly differentiable property of function within a scope of |e|≤δ. To guarantee the constantly differentiable property of function within the overall defined domains, the requirements given below must be satisfied:(22)$$\begin{array}{c} \left\{\begin{array}{c} newfal\left(\delta_{-},\alpha ,\delta \right)=newfal\left(\delta_{+},\alpha ,\delta \right)\\ newfal\left(-\delta_{-},\alpha ,\delta \right)=newfal\left(-\delta_{+},\alpha ,\delta \right) \end{array}\right. \end{array}$$
(23)$$\begin{array}{c} \left\{\begin{array}{c} newfal^{\prime }\left(\delta_{-},\alpha ,\delta \right)=newfal^{\prime }\left(\delta_{+},\alpha ,\delta \right)\\ newfal^{\prime }\left(-\delta_{-},\alpha ,\delta \right)=newfal^{\prime }\left(-\delta_{+},\alpha ,\delta \right) \end{array}\right. \end{array}$$

Putting Equations (Equation 20) and (Equation 21) into Equations (Equation 22) and (Equation 23), the coefficients in Equation (Equation 24) can be given as
(24)$$\begin{array}{c} \left\{\begin{array}{c} a_{1}=\frac{\alpha \delta^{\alpha -1}\left(\delta \cos\alpha \delta -\sin\alpha \delta \right)}{\alpha \arcsin\delta \cos\alpha \delta -\frac{\sin\alpha e}{\sqrt{1+\delta^{2}}}}\\ a_{2}=\frac{\delta^{\alpha -1}}{\cos\alpha \delta }-\frac{\delta^{\alpha -1}\left(\delta \cos\alpha \delta -\sin\alpha \delta \right)}{\sqrt{1+\delta^{2}}\alpha \arcsin\delta \cos^{2}\alpha \delta -\sin\alpha \delta \cos\alpha \delta } \end{array}\right. \end{array}$$

By replacing the coefficients of Equation (Equation 24) into Equation (Equation 21), the proposed novel nonlinear optimal control function is decided.

As can been seen in the above analysis for the optimal control function newfal(e,α,δ) design, its degree of difficulty of coefficients calculation can be accepted, and it can be differentiable and smooth at the segment point; in addition, parameter α can also be fully taken into account, so the proposed design is more reasonable and suitable than the primitive and traditional modified designs.

In this paper, ADRC using the optimal control function nfal(e,α,δ) and ADRC using the optimal control function newfal(e,α,δ) are abbreviated to NADRC and NewADRC, respectively.

### 4.2. Adaptive Nonlinear Decreasing Strategy for Convergence Factor

The key parameters of optimization algorithm have a certain degree of impact on its optimization performance. However, invariable, blind randomization or fixed change mode for parameters is not conducive to the global convergence of the algorithm. A large convergence factor for WOA should be selected in the early iteration so as to improve the global searchability of the algorithm. With the continuous evolution of population, a smaller convergence factor should be selected, which is conducive to the local searchability. An adaptive nonlinear decreasing strategy using the exponential form for the convergence factor is given in this paper.

The specific convergence factor calculation $a\left(t_{r}\right)$ function is as follows:(25)$$\begin{array}{c} a\left(t\right)=-0.5+2.5\times e^{{t_{r}}^{\beta_{a}}\times \ln\left(\frac{0.5}{2.5}\right)} \end{array}$$
where $\beta_{a}$ represents the adaptive nonlinear decreasing optimization factor for convergence factor; and $t_{r}$ represents the iteration progress, $t_{r}\;=\;t/T_{\max}$.

The diagram of the above adaptive nonlinear decreasing function for the convergence factor is shown in Figure 4.

As can be seen in Figure 4, if the adaptive nonlinear decreasing strategy for convergence factor is adopted, the convergence factor will be decreased nonlinearly from 2 to 0, and its deceleration rate changes with the iteration progress. Thus, the nonlinear decreasing trend curve for convergence factor can be optimized by choosing appropriate optimization factor $\beta_{a}$ so as to improve the global search capability ulteriorly.

### 4.3. Gaussian Mutation for Simulated Annealing

In traditional simulated annealing, average random distribution is a commonly used method for generating novel solutions in the neighborhood of the original individual. However, the difference-blind intensity for local search is not beneficial for generating superior solutions. Gauss distribution is a kind of commonly random distribution that obeys the law of normal distribution; the disturbance is called Gaussian disturbance, as its intensity obeys Gauss distribution. The Gaussian mutation mechanism is a local search mechanism by imposing Gaussian disturbance, and it can realize the mutation with constrained intensity [33]. According to the characteristics of normal distribution, compared with the mutation mechanism using average random distribution, Gaussian mutation can realize the key search of the local area nearby the original individual. Thus, through the introduction of Gaussian mutation, simulated annealing is improved to a considerable extent not only in the local search range and intensity but also in the escape possibility for lying in local minimum. A solution generating strategy for simulated annealing by introducing the Gaussian mutation mechanism is given in this paper.

The specific X(u) solution generating function by introducing the Gaussian mutation mechanism is as follows:(26)$$\begin{array}{c} X\left(u\right)=X_{e}\left(t,q\right)+\lambda (k-1)\cdot Gaussian\left(\mu ,\sigma^{2}\right) \end{array}$$
where λ(k−1) represent the weight vector of the disturbance characteristics for the elite set of the k−1-th period, $\lambda \left(k-1\right)=\frac{\sum_{w=1}^{NE}X_{e}\left(t,w\right)}{NE}$, NE represents the size of elite set $X_{e}$, and $Gaussian\left(\mu ,\sigma^{2}\right)$ is a Gaussian distribution random number with mean μ and standard deviation σ.

### 4.4. Learning Mechanism Using Mixtures of Competitive Mechanism and OBL Mechanism

Generally, the learning mechanism of the memetic algorithm is based on a competitive mechanism. However, the inferiority clustering caused by using a competitive mechanism poses a huge risk for population diversity in the later stages of evolution, and it leads to local convergence easily. The opposition-based learning (OBL) mechanism was proposed by Tizhoosh, which can generate massive superior opposition-based learning solutions far away from present local optimal solution, and it is helpful for improving the global convergence capability [34].

The specific opposition-based learning formula is as follows:(27)$$\begin{array}{c} \left\{\begin{array}{c} {X_{e}{\left(t,q\right)}^{\prime }}_{i}=k\left(a_{i}+b_{i}\right)-X_{e}{\left(t,q\right)}_{i}\\ {X_{e}{\left(t,q\right)}^{\prime }}_{i},{X_{e}\left(t,q\right)}_{i}\in \left[a_{i},b_{i}\right]\\ i\in \left[1,2,\ldots ,d\right] \end{array}\right. \end{array}$$
where $a_{i}$ and $b_{i}$ represent the minimum and maximum values on the boundary of the *i*-th dimension; $k\in \left[0,1\right]$ is the random generalization coefficient; ${X_{e}{\left(t,q\right)}^{\prime }}_{i}$ represents the *i*-th dimension of the opposition-based learning solution for the q-th elite individual; and *d* represents the number of solution dimension.

In the iteration process, a proportion of solutions is likely to fall into ‘overflow’, and these ‘overflow’ solutions are necessary to deal with immediately; otherwise, the optimize performance of the memetic algorithm will be weakened.

The specific overflow disposal formula in this paper is as follows:(28)$$\begin{array}{c} x{{\left(t\right)}^{\prime }}_{i}=a_{i}+\beta \left(b_{i}-x{\left(t\right)}_{i}\right) \end{array}$$
where $\beta \in \left[0,1\right]$ is the random overflow disposal coefficient. The search in the WOA algorithm depends entirely on randomness, resulting in low convergence accuracy and slow convergence speed. Therefore, an improved WOA on the basis of chaotic sequences and adaptive cross-mutation is developed in this paper, which greatly improves convergence speed and precision.

### 4.5. Elite Set Maintenance Mechanism Based on Fusion Distance

In the evolution process of each iteration, the high-quality solutions in the population will be expanded into an elite set so as to save the existing optimization achievements effectively. Aiming at preventing an adverse effect for algorithm computational efficiency about the rapid growth of the elite set size, elite set size $E_{s}$ should be less than the allowable maximum elite set size $E_{a}s$, that is, $E_{S}\leq E_{AS}$. So, the design of the elite set maintenance mechanism is very significant for MA. The distance measurement is an important part of elite set maintenance. The Euclidean distance is popularly used in traditional optimization algorithms for elite set maintenance; these redundant and crowded solutions with shorter Euclidean distance will be deleted. However, the calculation of Euclidean distance is dependent on the dimensions of variables; in addition, the linear distance between solutions is calculated by Euclidean distance, but because the distribution of solutions is not considered, the Euclidean distance cannot measure the correlation between variables [35]. Similarly, the Mahalanobis distance is also an accurate distance measurement. Based on these considerations, the fusion distance combined linear-weighted total value of the Mahalanobis distance and Euclidean distance is given in this paper.

The specific calculated formula in this paper about fusion distance $d_{Mix}$ is as follows:(29)$$\left\{\begin{array}{c} d_{Mix}=\omega \times MD\left(X,Y\right)+\left(1-\omega \right)\times ED\left(X,Y\right)\\ \begin{array}{c} C_{Y}=\left[\begin{array}{c} \begin{array}{c} \;\rho_{Y_{1}Y_{1}} \end{array}\;\rho_{Y_{1}Y_{2}}\cdots \rho_{Y_{1}Y_{n}}\\ \rho_{Y_{2}Y_{1}}\rho_{Y_{2}Y_{2}}\cdots \rho_{Y_{2}Y_{n}}\\ \;\;\vdots \;\;\vdots \;\;\ddots \;\;\vdots \\ \rho_{Y_{n}Y_{{}_{1}}}\rho_{Y_{n}Y_{2}}\cdots \rho_{Y_{n}Y_{n}} \end{array}\right]\\ \omega =\sqrt{1-\left|C_{Y}\right|} \end{array} \end{array}\right.$$
where MD represents the Mahalanobis distance, $C_{Y}$ represents the correlation coefficient matrix for the sample set *Y*, *n* represents the sample set size *Y*, $Y_{i}\left(i=1,\ldots ,n\right)$ represents the corresponding elements for sample set *Y*, ρ represents the correlation coefficient, ω represents the weight value about relevant information of the Mahalanobis distance, and the other 1−ω is the weight value about relevant information of the Euclidean distance [36].

### 4.6. Design of ADRC Based on Improved Memetic Algorithm

The design core of the ADRC based on improved memetic algorithm (IMA) is the integration parameters optimization mechanism for ADRC. Aiming at improving the integration parameters optimization effect, an adaptive nonlinear decreasing strategy for the convergence factor, Gaussian mutation mechanism, improved learning mechanism and an elite set maintenance mechanism based on fusion distance are integrated into the moth–flame algorithm.

The flowchart for the proposed ADRC based on improved memetic algorithm is shown in Figure 5.

## 5. Experiment Analysis

### 5.1. Experiment Platform

In order to better verify the proposed active disturbance resisting controller (ADRC) based on IMA, an experiment platform is adopted, which includes the actual controllers, PMSM, inverter, sensor, etc. The real-time resistance is provided by using PMSM dynamic loading, and reference speed and torque signal flow for PMSM velocity control is set in advance so as to establish the corresponding virtual PMSM velocity control environment. The physical system diagram of the experimental platform is shown in Figure 6.

In Figure 6, such as velocity control PMSM, dynamic loading PMSM, velocity control controller, dynamic loading controller, intelligent digital torque/speed sensor and its measuring instrument, monitoring computer, transformer, main breaker and RS485 serial transmission lines are significant components for experiment platform. The RS485 serial transmission lines can connect with the monitoring computer and controller. The PMSM velocity controller is suitable for the velocity reference curve with uniform, smooth slope line increase or decrease and finite-amplitude sinusoidal disturbances.

The PMSM velocity control environment in the experiment is built with real PMSM and velocity controller containing DSP chip. A detailed configuration of the experiment platform is shown below: dynamic loading PMSM and velocity control PMSM share identical parameters, and the corresponding rated voltage, current, power, velocity and torque are set as 220 V, 4.18 A, 750 W, 3000 r/min and 2.39 Nm. In addition, the corresponding safety overload rate of torque is set as 0.83, in other words, the short-term safety torque is 4.37 Nm. Configuration in the dynamic loading monitoring computer as well as speed control monitoring computer remains identical, and the relevant processor is ‘Core i7-7700K @ 4.2GHZ’. Monitoring software revision is ‘Visual studio 2017’. Controller core chip and programming software revisions belong to ‘TMS320F28335’ and ‘CCS 6.0’. Embedded LCD display screen type for control circuit board belongs to ‘12864B V2.0’. The type of intelligent digital torque and speed sensor as well as matching measure instruments are ‘NO. JN338’. The velocity and torque ranges reach 0–6000 r/min and 0–20 Nm.

### 5.2. Experiment Scenario

In this paper, as far as the modified NewADRC redesigned in this paper, the optimization integration of ADRC key parameters is obtained by IMA, and the specific ADRC is abbreviated as NewADRC-IMA. Detailed parameters of improved IMA are shown below: the population size is set as 40, the maximum number of iteration times is set as 80, the probability of the surrounding preys is set as 0.6, the adaptive nonlinear decreasing optimization factor for convergence factor is set as 1.75, the probability of mutation selection behavior is set as 0.15, the initial temperature is 200 °C, the cooling coefficient is 0.75, the termination temperature is 50 °C, and the elite size is set as 25. Detailed results about the optimization integration of ADRC key parameters by using IMA are shown below: obtained optimization integrated parameters $\alpha_{11}=0.90$, $\alpha_{12}=0.53$, $\omega_{0}=357.7$, $\alpha_{21}=0.63$, $\alpha_{22}=0.35$ and $\omega_{c}=26.0$.

In order to verify the performance of NewADRC-IMA proposed in this paper, PMSM velocity control experiments are implemented. Fuzzy PI and traditional ADRC based on the genetic algorithm are two popular traditional control algorithms, widely used in PMSM velocity control due to the traits of being stable in control and easily realized; however, their tracking control accuracy is relatively low. In this paper, traditional ADRC based on the genetic algorithm is abbreviated as ADRC-GA. An improved ADRC with effective strategies was proposed to solve the defect that the nonlinear function of traditional ADRC is not differentiable at the piecewise point, and an improved moth–flame optimization (MFO) was proposed to obtain its key parameters, which can effectively improve the tracking control performance. The above specific modified ADRC is abbreviated as NADRC-IMFO.

The PMSM velocity control scenario involved in the research can be given as below: the experimental time is set as 0.4 s; the reference speed is set as 1000 r/min, and beginning with upload; the experimental uploaded torque is set as 0.05 Nm; and the experimental load torque is set as 2 Nm. The boundary constraints for the PMSM velocity control scenario are shown as follows: the maximum acceptable stable control speed error $Amax(SE_{S})$ and torque error $Amax(SE_{T})$ are set as 2.5 r/min and 0.8 Nm, and the maximum acceptable speed error $Amax(E_{S})$ and torque error $Amax(E_{T})$ are set as 250 r/min and 5.5 Nm. In addition, the reference speed and torque signal flow for PMSM velocity control are necessary to be set up. The reference speed and torque signal flow diagram of experiment platform is shown in Figure 7.

The automatic capture configuration is prescribed as follows: The automatic capture curves can be classified into torque and speed types. The control algorithms include fuzzy PI, ADRC-GA, NADRC-IMFO and NewADRC-IMA (proposed in this paper). The dynamic loading torque signals total 0.053 Nm and 2.153 Nm under sequential and cycle transmission types. The torque transmission efficiency totals 0.944 and 0.952 at 0 r/min and 1000 r/min point. As shown in Figure 7, the time points in preparation, rotating speed, loading, and capture-start reach 0 s, 0.8 s, 1.1 s and 0.8 s; the time width in preparation, rotating speed, loading, and capture reach 0.8 s, 1.2 s, 0.9 s and 0.6 s; and the cycle number and total time reach 8 and 16 s.

In view of conventional use and emergency (or escape) treatment, the stable speed-regulating and quick speed-regulating modes about the PMSM velocity control can be chosen in practice. In quick speed-regulating mode, rapid start-up and achieved reference are necessary; contrarily, in quick speed-regulating mode, smooth transition is the most important consideration. In order to verify the effectiveness of the IADRC in this paper, two practical cases are given. The first practical case is set up for reflecting quick speed-regulating mode, and there are no additional alterations based on the above PMSM velocity control scenario. The second practical case is set up for reflecting the stable speed-regulating mode, where enough time should be given for the start-up and load operation, and for which the specific configurations are as follow: the start-up and load operation types follow ’slick inclined line’, the start-up and load operation duration time are set as 0.01s.

### 5.3. Experiment Result and Analysis

The performance of proposed control algorithm NewADRC-IMA is necessary to be verified, and three control algorithms (fuzzy PI, ADRC-GA and NADRC-IMFO) are used for comparison.

The experiment results and analyses of PMSM velocity control experiment are provided as follows.

Table 1, Table 2, Table 3 and Table 4 list the speed and torque ITAE for two PMSM speed control practical cases.

As listed in Table 1, Table 2, Table 3 and Table 4, compared with fuzzy PI, ADRC-GA or NADRC-IMFO, except $\left|E_{T}max\right|$, the NewADRC-IMA can obtain lesser $ITAE_{S}$, $ITAE_{T}$, $\left|E_{S}max\right|$, $\left|SE_{S}max\right|$, $\left|SE_{T}max\right|$ and At in the PMSM velocity control practical cases, and there are very significant decreasing degrees.

The specific speed and torque curves gained under the two PMSM speed control practical cases can be seen from Figure 8, Figure 9, Figure 10 and Figure 11.

As shown in Figure 8 and Figure 10, compared with the speed curves for the tracking trajectory using fuzzy PI, ADRC-GA or NADRC-IMFO, the speed fluctuation degree can be reduced, and the restricting ability for overshoot can be improved by using NewADRC-IMA effectively. As shown in Figure 9 and Figure 11, compared with torque curves for the tracking trajectory using fuzzy PI, ADRC-GA or NADRC-IMFO, the torque ripple can be descended by using the NewADRC-IMA effectively.

Clearly, the improved strategies introduced in this paper are effective. The nonlinear optimal control function constructed in this paper is differentiable and smooth at the piecewise point for modifying ADRC, and IMA proposed in this paper has powerful global optimization capability, so the modified ADRC based on IMA is improved obviously.

## 6. Conclusions

For the complex practical PMSM speed control problem, the key parameters of control algorithm have a noteworthy influence on the control effect. Furthermore, the optimal control function are also a non-negligible issue for it.

A modified ADRC of PMSM based on an IMA is designed in this paper. Specifically, for solving the issue that the traditional optimal control function for ADRC is non-differentiable and unsmoothed at the piecewise point, a modified novel differentiable and smooth nonlinear optimal control function is constructed. Furthermore, an improved memetic algorithm based on an adaptive nonlinear decreasing convergence factor strategy for the whale optimization algorithm, a Gaussian mutation for simulated annealing, a learning mechanism using mixtures of competitive mechanism and OBL mechanism and an elite set maintenance mechanism based on fusion distance is proposed for solving the issue that the traditional intelligent optimization algorithm cannot optimally integrate key parameters of ADRC effectively. Compared with fuzzy PI, ADRC-GA, NADRC-IMFO, the modified ADRC of PMSM based on the IMA proposed in this paper has better tracking control performance.

In order to verify the performance of NewADRC-IMA proposed in this paper effectively, an experimental platform of PMSM velocity control was built. The experimental results can illustrate the efficacy of the NewADRC-IMA. Compared with the existing representative control algorithms (PI, ADRC-GA, and NADRC-IMFO), the NewADRC-IMA has several significant performance advantages, such as faster response, smaller steady-state error, tinier overshoot, etc.

There are several future directions suggestions for ADRC of PMSM: I draw lessons from the existing advanced control methods, such as robust predictive control, fuzzy predictive control or adaptive model predictive control, to further improve its control performance; II design a more suitable novel nonlinear optimal control function so as to improve its control accuracy; III combine the improved ADRC with fuzzy controller to further improve its control performance; and IV construct a verification environment using the complex experiment so as to further improve the verification precision. 

## Figures and Tables

**Figure 1 sensors-23-03621-f001:**
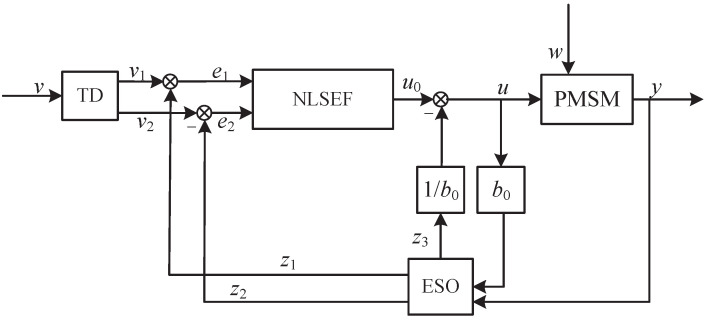
Control structure diagram about the active disturbance rejection controller.

**Figure 2 sensors-23-03621-f002:**
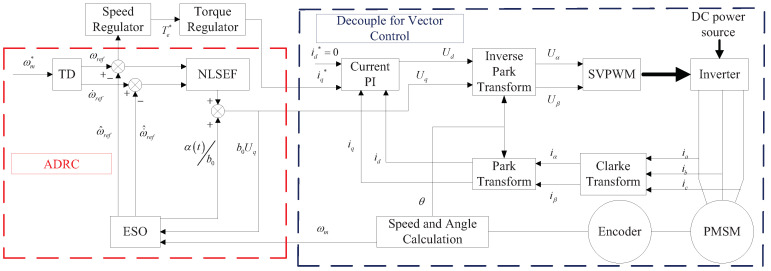
Control block diagram of PMSM speed control system based on ADRC.

**Figure 3 sensors-23-03621-f003:**
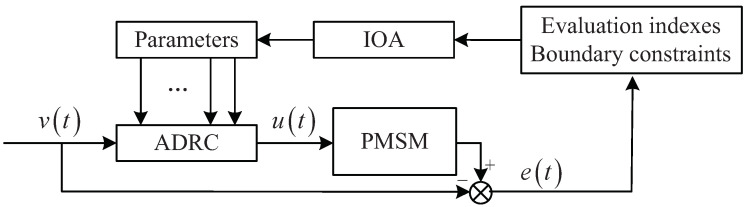
Diagram about the ADRC parameters adjusting method using intelligent optimization algorithm.

**Figure 4 sensors-23-03621-f004:**
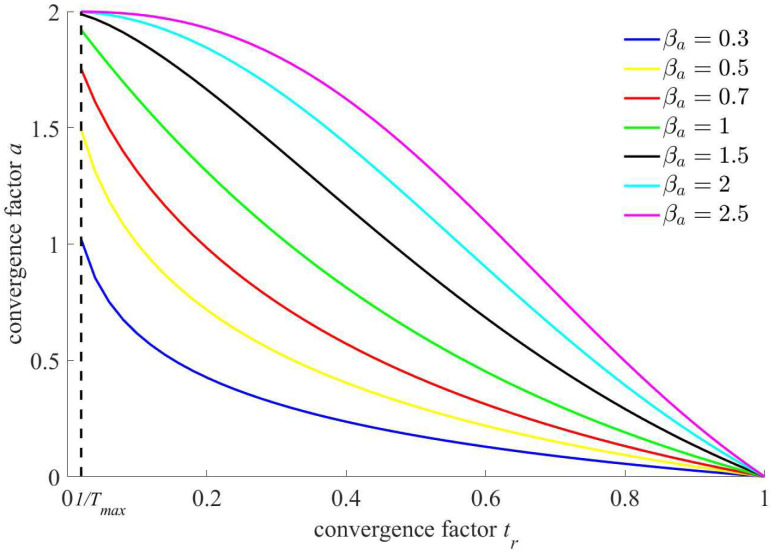
Diagram of adaptive nonlinear decreasing function for convergence factor.

**Figure 5 sensors-23-03621-f005:**
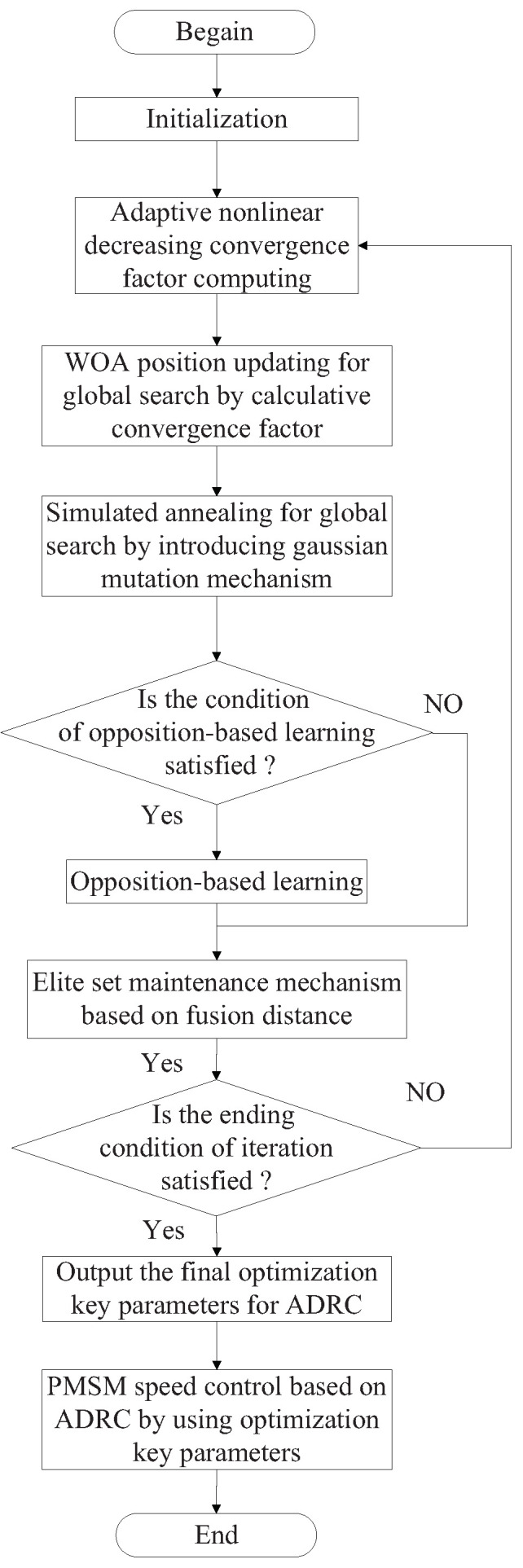
Flowchart of ADRC based on improved memetic algorithm.

**Figure 6 sensors-23-03621-f006:**
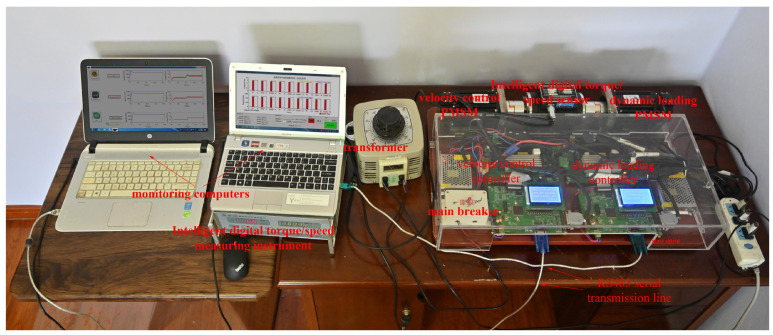
The physical system diagram of experiment platform.

**Figure 7 sensors-23-03621-f007:**
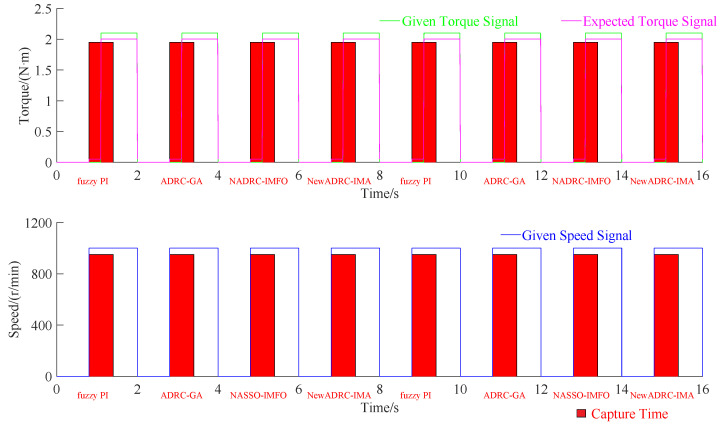
The reference speed and torque signal flow diagram of experiment platform.

**Figure 8 sensors-23-03621-f008:**
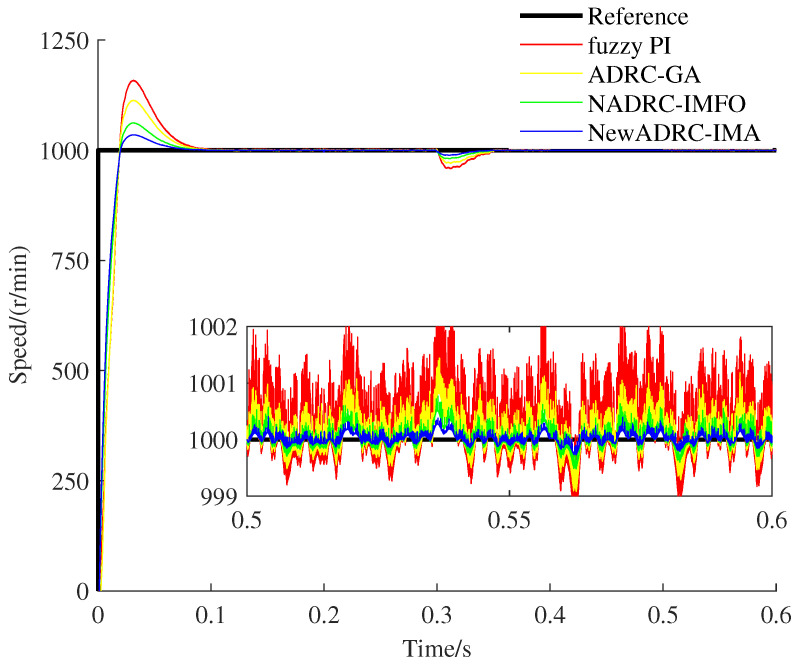
The speed curves under the first PMSM velocity control practical case.

**Figure 9 sensors-23-03621-f009:**
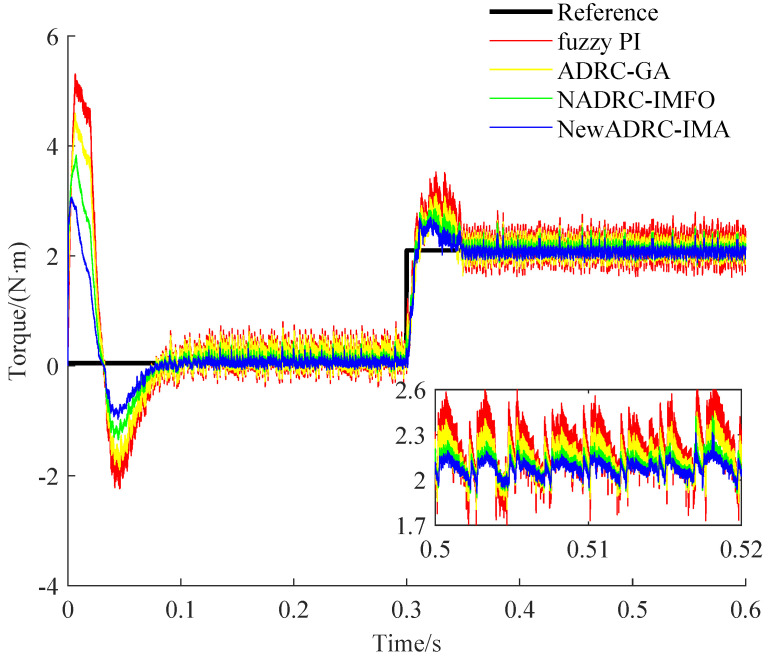
The torque curves under the first PMSM velocity control practical case.

**Figure 10 sensors-23-03621-f010:**
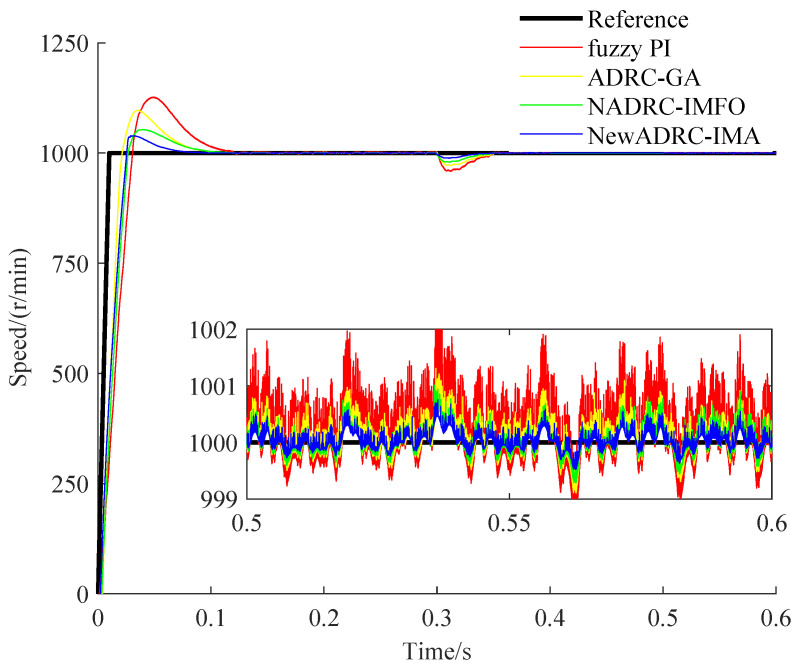
The speed curves under the second PMSM velocity control practical case.

**Figure 11 sensors-23-03621-f011:**
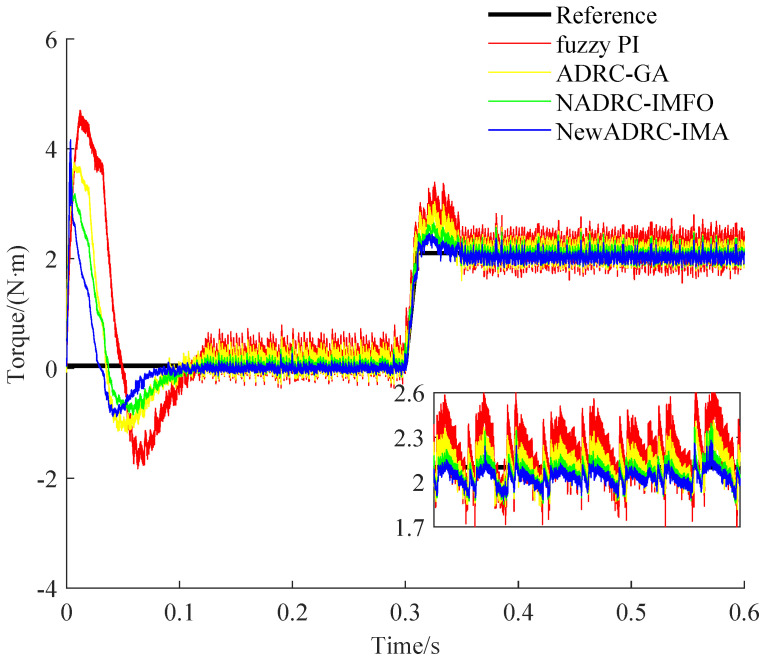
The torque curves under the second PMSM velocity control practical case.

**Table 1 sensors-23-03621-t001:** The speed and torque ITAE for each practical case.

Variable	Practical Case	PI	ADRC-GA	NADRC-IMFO	NewADRC-IMA
$ITAE_{S}$	first practical case	$7.95\times 10^{-1}$	$4.67\times 10^{-1}$	$1.25\times 10^{-1}$	$8.64\times 10^{-2}$
$ITAE_{S}$	second practical case	$6.61\times 10^{-1}$	$4.30\times 10^{-1}$	$9.35\times 10^{-2}$	$5.80\times 10^{-2}$
$ITAE_{T}$	first practical case	$8.27\times 10^{-2}$	$3.26\times 10^{-2}$	$7.35\times 10^{-3}$	$3.81\times 10^{-3}$
$ITAE_{T}$	second practical case	$5.31\times 10^{-1}$	$2.44\times 10^{-1}$	$6.65\times 10^{-2}$	$3.69\times 10^{-3}$

**Table 2 sensors-23-03621-t002:** The maximum absolute value of speed and torque error for each practical case.

Variable	Practical Case	PI	ADRC-GA	NADRC-IMFO	NewADRC-IMA
$\left|E_{S}max\right|$	first practical case	1158 r/min	1113 r/min	1061 r/min	1037 r/min
$\left|E_{S}max\right|$	second practical case	1127 r/min	1095 r/min	1056 r/min	1040 r/min
$\left|E_{T}max\right|$	first practical case	5.26 Nm	4.58 Nm	3.72 Nm	3.09 Nm
$\left|E_{T}max\right|$	second practical case	4.70 Nm	3.85 Nm	3.77 Nm	4.13 Nm

**Table 3 sensors-23-03621-t003:** The maximum absolute value of stable control speed and torque error for each practical case.

Variable	Practical Case	PI	ADRC-GA	NADRC-IMFO	NewADRC-IMA
$\left|SE_{S}max\right|$	first practical case	2.47 r/min	1.58 r/min	0.87 r/min	0.76 r/min
$\left|SE_{S}max\right|$	second practical case	2.34 r/min	1.27 r/min	0.96 r/min	0.90 r/min
$\left|SE_{T}max\right|$	first practical case	0.75 Nm	0.64 Nm	0.57 Nm	0.44 Nm
$\left|SE_{T}max\right|$	second practical case	0.72 Nm	0.63 Nm	0.59 Nm	0.37 Nm

**Table 4 sensors-23-03621-t004:** The adjusting time for each practical case.

Variable	Practical Case	PI	ADRC-GA	NADRC-IMFO	NewADRC-IMA
At	first practical case	0.090 s	0.087 s	0.083 s	0.076 s
At	second practical case	0.102 s	0.093 s	0.094 s	0.081 s

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
