# Peer review of "Modified ADRC Design of Permanent Magnet Synchronous Motor Based on Improved Memetic Algorithm"

_sensors, 2023, doi:10.3390/s23073621_

Round 1
Author Response
We sincerely thank you for the comments that helped a lot to improve the quality of our submission.
- Some formula symbols, such as Rs in (1), are not explained clearly. In addition, dq in symbol interpretation is not reflected in the formula, etc.
Response:
- Thanks for your constructive comment to improve our manuscript!
- In the revised version, we have rewritten the abstract to show clearly the objective of this paper as follows.
In this paper, a modified novel auto disturbance rejection control (ADRC) design of permanent magnet synchronous motor based on improved memetic algorithm (IMA) is proposed. Firstly, there is obvious system ripple caused by the defect that optimal control function used in traditional ADRC can't be differentiable and smooth at the segment point, aiming at weakening the system ripple effectively, the proposed method constructs a novel differentiable and smooth optimal control function to modify ADRC design. Furthermore, aiming at improving the integration parameters optimization effect effectively, a novel improved memetic algorithm is proposed for obtaining optimal parameters of ADRC. Specifically, an IMA with the high-quality balance based on adaptive nonlinear decreasing strategy for convergence factor, gaussian mutation mechanism, improved learning mechanism with the high-quality balance between competitive and opposition-based learning (OBL) and an elite set maintenance mechanism based on fusion distance is proposed, so that these strategies can improve optimization precision by a large margin. Finally, the experiment results of PMSM speed control practical cases show that the modified ADRC based on IMA has an apparent better optimization effect than that of fuzzy PI, traditional ADRC based on genetic algorithm and an improved ADRC based on improved moth-flame optimization.
- The authors should compare the performance of the proposed control strategy with other methods.
Response:
- Thanks for your constructive comment to improve our manuscript!
- In the revised version, the evidence of comparisons and analysis for comparing performance of the proposed control strategy (modified ADRC based on IMA) with other methods (fuzzy PI, traditional ADRC based on genetic algorithm and improved ADRC with effective strategies) has been provided as follows.
Finally, the experiment results of PMSM speed control practical cases show that the modified ADRC based on IMA has an apparent better optimization effect than that of fuzzy PI, traditional ADRC based on genetic algorithm and an improved ADRC based on improved moth-flame optimization.
- The authors want to present the contribution and novelty of this paper.
Response:
- Thanks for your constructive comment to improve our manuscript!
- In the revised version, the contribution and novelty have been provided as follows.
As for the problem that the existing active disturbance rejection controllers of PMSM controllers do not have sufficient competence, this paper proposes a novel modified ADRC of PMSM based on improved memetic algorithm (IMA). The following summarizes the main contributions of this paper:
(Ⅰ) An improved memetic algorithm (IMA) based on an adaptive nonlinear decreasing convergence factor strategy for whale optimization algorithm, a Gaussian mutation for simulated annealing, a learning mechanism using mixtures of competitive mechanism and OBL mechanism and an elite set maintenance mechanism based on fusion distance are proposed.
(Ⅱ) A novel differentiable and smooth nonlinear function is constructed for modifying ADRC, for overcoming the non-differentiable and unsmoothed situation at the piecewise point of nonlinear functions in the conventional ADRC; the above IMA is proposed for improving the effectiveness for the optimization integration of ADRC key parameters, so as to improve the performance for modified ADRC of PMSM based on IMA effectively.
For the proposed novel modified ADRC of PMSM based on improved memetic algorithm (IMA), the following summarizes the major novelties of this paper:
(Ⅰ) An improved memetic algorithm (IMA) with the high-quality global optimization performance is proposed. In this proposed algorithm, an improved whale optimization algorithm (IWOA) by adopting an adaptive nonlinear decreasing strategy for convergence factor is designed for global search, an improved simulated annealing (ISA) by introducing gaussian mutation mechanism is designed for local development, an improved learning mechanism between elite set and common individual population based on competitive and opposition-based learning and an elite set maintenance mechanism based on fusion distance and congestion degree distance are incorporated for improving global optimization performance effectively.
(Ⅱ) Based on the traditional optimal control function, inverse hyperbolic sine function, sine function are introduced, and a novel optimal control function is constructed by fitting method for modifying ADRC.
- Presentation need to be revised carefully.
Response:
- Thanks for your constructive comment to improve our manuscript!
- In the revised version, we have revised the reference as follows.
Gashti, A., Akbarimajd, A. Designing anti-windup PI controller for LFC of nonlinear power system combined with DSTS of nuclear power plant and HVDC link. Electrical Engineering 2020, 102(2), 793-809.
Sira-Ramírez, H., Linares-Flores, J., García-Rodríguez, C., Contreras-Ordaz, M. On the Control of the Permanent Magnet Synchronous Motor: An Active Disturbance Rejection Control Approach. IEEE Transactions on Control Systems Technology 2014, 22(5), 2056-2063.
Tolba, M., Diab A., Tulsky, V., Tulsky, V., Abdelaziz, A. LVCI approach for optimal allocation of distributed generations and capacitor banks in distribution grids based on moth-flame optimization algorithm. Electrical Engineering 2018, 100(3), 2059-2084.
Xu, L.; Li, Y.; Li, K.; Beng, G. H.; Jiang, Z.; Wang, C.; Liu, N. Enhanced Moth-flame Optimization Based on Cultural Learning and Gaussian Mutation. Journal of Bionic Engineering 2018, 15(4), 751-763.
- The abstract of the article is very long.
Response:
- Thanks for your constructive comment to improve our manuscript!
- In the revised version, we have rewritten the abstract to show clearly the objective of this paper as follows.
In this paper, a modified novel auto disturbance rejection control (ADRC) design of permanent magnet synchronous motor based on improved memetic algorithm (IMA) is proposed. Firstly, there is obvious system ripple caused by the defect that optimal control function used in traditional ADRC can't be differentiable and smooth at the segment point, aiming at weakening the system ripple effectively, the proposed method constructs a novel differentiable and smooth optimal control function to modify ADRC design. Furthermore, aiming at improving the integration parameters optimization effect effectively, a novel improved memetic algorithm is proposed for obtaining optimal parameters of ADRC. Specifically, an IMA with the high-quality balance based on adaptive nonlinear decreasing strategy for convergence factor, gaussian mutation mechanism, improved learning mechanism with the high-quality balance between competitive and opposition-based learning (OBL) and an elite set maintenance mechanism based on fusion distance is proposed, so that these strategies can improve optimization precision by a large margin. Finally, the experiment results of PMSM speed control practical cases show that the ADRC based on IMA has an apparent better optimization effect than that of fuzzy PI, traditional ADRC based on genetic algorithm and an improved ADRC based on improved moth-flame optimization.
- Function ??????(?,?, ?) in equation (9) must be odd, so term ?? ? is excess. Also, ? has no role in this equation.
Response:
- Thanks for your constructive comment to improve our manuscript!
- In the revised version, the novel optimal control function ??????(?,?, ?) has been redesigned, the novel optimal control function formula with is odd, and ? has role in this equation.
- In the revised version, we have redesigned the novel optimal control function ??????(?,?, ?) of this paper as follows.
As to the issues about the non-smooth linear segment and the non-differentiable at the piecewise point in function in case of , it is replaced by linear function containing inverse hyperbolic function with superior smoothness and sin function in this paper.
The specific formula about the novel optimal control function with is as follows:
The specific formula about the novel optimal control function with is as follows:
If , the optimal control function comprises inverse hyperbolic sine function, quadratic function and sine function, and it is capable to guarantee the constantly differentiable property of function within a scope of . To guarantee the constantly differentiable property of function within the overall defined domains, the requirements given below must be satisfied:
Putting equation (20) and equation (21) into equation (22) and equation (23), then the coefficients in equation (24) can be given as:
By replacing the coefficients of equation (24) into equation (21), the proposed novel nonlinear optimal control function is decided.
As can been seen the above analyze for the optimal control function design, its degree of difficulty of coefficients calculation can be accepted, and it can be differentiable and smooth at the segment point, in addition, parameter can also be fully taken into account, so the proposed design is more reasonable and suitable than the primitive and traditional modified designs.
- The amount of chattering of the torque function is very high, which is not acceptable.
Response:
- Thanks for your constructive comment to improve our manuscript!
- There are various factors about the amount of chattering of the torque. In order to better describe it, please allow me to give several definitions as follow.
During PMSM speed tracking control process, the time about the long-term control range with the acceptable maximum absolute value of stable control speed and torque error ( and ) is called as stable time , the above referred error should be less than acceptable maximum stable control speed and torque error ( and ); the time between the begin time point of power-up and the corresponding begin time point of stable control range is called as adjusting time , adjusting time must not exceed maximum acceptable adjusting time ; the maximum absolute value of speed and torque error ( and ) should be less than maximum acceptable control speed and torque error ( and ).
- Firstly, please allow me to give the reason for the amount of chattering of the maximum absolute value of torque error is high. The PMSM speed tracking control is a sophisticated problem which is multiple objectives, large delay and nonlinearity. In fact, due to uploaded torque is only 0.05 Nm, the maximum absolute value of torque error is equal to the maximum torque during whole PMSM speed tracking control process. In order to reach reference speed quickly, the torque must be large enough, and the amount is related to adjusting time . According to the specific calculated formula for the integral of time multiplied by the absolute value of error, , adjusting time and the acceptable maximum absolute value of torque error must be set, in theory, if they are not set, adjusting time will be close to 0, and the acceptable maximum absolute value of torque error will become infinite. Aiming at obtaining a sufficiently small adjusting time , in the original version, the maximum acceptable adjusting time is set as 0.065s, and the acceptable maximum absolute value of torque error is set as 8N·m. This setup is a little severe for the specific PMSM in this paper. The rated torque and safety overload rate of torque for the specific PMSM in this paper have set as 2.39 N·m and 83%, so the PMSM can be continuous used normally more than 3 hour (1.08×104s) under 4.37 N·m (1.83×2.39 N·m) condition, however, 8N·m is approximately equivalent to 3.3×2.39. So, the maximum acceptable adjusting time and the acceptable maximum absolute value of torque error should be reduced appropriately.
- Further more, please allow me to give the reason for the amount of chattering of maximum absolute value of stable torque error is high. In the original version, PI and traditional ADRC with experienced parameters are chosen as comparison algorithms, PI and traditional ADRC with experienced parameters are not high-accuracy control algorithms for the specific PMSM speed control practical cases in this paper. In addition, in the original version, the fitness value is linear weighted value for speed and torque, and speed and torque weight are set as 0.7 and 0.3, the torque weight is a little small.
- In the revised version, we have reset the maximum acceptable adjusting time and the acceptable maximum absolute value of torque error as appropriate value (0.105s and 5.5 Nm) in engineering. In engineering, the acceptable maximum absolute value of torque error are close to the critical value of short-term safety torque or a little large, so as to quick response and enter into the stable range timely. In this paper, the short-term safety torque is 4.37 N·m (1.83×2.39 N·m). In the revised version, fuzzy PI and traditional ADRC based on genetic algorithm are chosen as comparison algorithms, and the significance of speed and torque are equal.
- In the revised version, the experiment results of PMSM velocity control experiment are provided in section 5.3.
- The amount of innovation of the article is low. It is not acceptable to define only one new function ??????(?,?, ?) and not analyze its properties. In addition, many such functions can be defined, which may have better performance.
Response:
- Thanks for your constructive comment to improve our manuscript!
- In the revised version, the analyze about both necessary and comparative advantage are provided, the section 2.2 has been rewritten partially, and section 4.1 has been rewritten partially too, suggestions as future directions has been added partially.
- In the revised version, we have given analyze and suggestions as future directions for the novel optimal control function ??????(?,?, ?) of this paper as follows.
The specific calculation formula of traditional optimal control function is as follows:
And the differential form of the derivative function for optimal control function is as follows:
At the segment point , the value of the function is:
When the function is differentiable in the segment, restraint condition is should be met. In this case, parameter , and function , led to destroying the nonlinear characteristics of the optimal control function. Thus, redesigning the optimal control function is clearly necessary.
As to the issues about the non-smooth linear segment and the non-differentiable at the piecewise point in function in case of , it is replaced by linear function containing inverse hyperbolic function with superior smoothness and sin function in this paper.
The specific formula about the novel optimal control function with is as follows:
The specific formula about the novel optimal control function with is as follows:
If , the optimal control function comprises inverse hyperbolic sine function, quadratic function and sine function, and it is capable to guarantee the constantly differentiable property of function within a scope of . To guarantee the constantly differentiable property of function within the overall defined domains, the requirements given below must be satisfied:
Putting equation (20) and equation (21) into equation (22) and equation (23), then the coefficients in equation (24) can be given as:
By replacing the coefficients of equation (24) into equation (21), the proposed novel nonlinear optimal control function is decided.
As can been seen the above analyze for the optimal control function design, its degree of difficulty of coefficients calculation can be accepted, and it can be differentiable and smooth at the segment point, in addition, parameter can also be fully taken into account, so the proposed design is more reasonable and suitable than the primitive and traditional modified designs.
There are several future directions suggestions for ADRC of PMSM: Ⅰ draw lessons from the existing advanced control methods, such as robust predictive control, fuzzy predictive control or adaptive model predictive control, to further improve its control performance; Ⅱ designing a more suitable novel nonlinear optimal control function, so as to improve its control accuracy; Ⅲ combine the improved ADRC with fuzzy controller, to further improve its control performance Ⅳ construct a verification environment using the complex experiment so as to further improve the verification precision.
- Finally, the comparison should be made based on several criteria and compared with different articles.
Response:
- Thanks for your constructive comment to improve our manuscript!
The response are as the same as the response about comment 7.

Reviewer 2 Report
The auto disturbance rejection control (ADRC) design of the permanent magnet synchronous motor based on a memetic algorithm is suggested in this research to successfully improve the speed control quality of the permanent magnet synchronous motor (PMSM). There was little control impact with the earlier control techniques of PI or conventional ADRC for PMSM. This research presents a revolutionary ADRC PMSM design for speed control based on a memetic algorithm. The technique uses a unique differentiable and smooth optimum control function to enhance ADRC for PMSM. The paper generally sounds good, but needs some revisions:
-the introduction is weak and need more references with detailed info
-add a diagram in section 2 to better see the problem
-add more details into control block diagram; the main equations
-waht is the motivation of using of memetic algorithm?
-there is no evidence to use to show the superiority of the suggested algorithm; add some comparisons and analysis with new methods
-how the method can be improved using fuzzy controllers such as:
Modeling renewable energy systems by a self-evolving nonlinear consequent part recurrent type-2 fuzzy system for power prediction
Author Response
We sincerely thank you for the comments that helped a lot to improve the quality of our submission.
Overall evaluation: The auto disturbance rejection control (ADRC) design of the permanent magnet synchronous motor based on a memetic algorithm is suggested in this research to successfully improve the speed control quality of the permanent magnet synchronous motor (PMSM). There was little control impact with the earlier control techniques of PI or conventional ADRC for PMSM. This research presents a revolutionary ADRC PMSM design for speed control based on a memetic algorithm. The technique uses a unique differentiable and smooth optimum control function to enhance ADRC for PMSM. The paper generally sounds good, but needs some revisions.
Response:
- Thanks for your overall evaluation!
- The introduction is weak and need more references with detailed info.
Response:
- Thanks for your constructive comment to improve our manuscript!
- In the revised version, we have rewritten the introduction to show more references with detailed information of this paper as follows.
An active disturbance rejection controller (ADRC) based on deep reinforcement learning (DRL) algorithm was proposed to be used in the flux weakening control (FWC) system of motors for more electric aircraft in [9]. A class of linear-nonlinear switching active disturbance rejection control (ADRC) to design speed controllers and current controllers for permanent magnet synchronous machine (PMSM) in servo systems was investigated in [10]. The Active disturbance rejection control (ADRC) and feedback compensation control method that can solve the speed fluctuation problem of permanent magnet synchronous motors was proposed in [11].
A memetic evolutionary multi-objective optimization method for power unit commitment problem was proposed in [16].
An improved whale optimization algorithm based on the Tchebycheff decomposition method, convergence factor nonlinear decline strategy, and genetic evolution measurement for model predictive controller was proposed in [18].
- Add a diagram in section 2 to better see the problem.
Response:
- Thanks for your constructive comment to improve our manuscript!
- In the revised version, we have added the control block diagram of PMSM speed control system based on ADRC as follows.
- Add more details (the main equations) into control block diagram.
Response:
- Thanks for your constructive comment to improve our manuscript!
- In the revised version, more details (the main equations) about control block diagram have been provided as follows.
The PMSM speed control system based on ADRC is composed of an ADRC, an inverter circuit and a PMSM. The ADRC controls the speed and torque of the PMSM. The specific meaning of the input and output signals of each module in the PMSM speed control system needs to be determined according to the controlled object. The transformation equation about PMSM of equations (2) to (4) is as follows:
The greatest advantage of ADRC is its capability in estimating disturbances inside and outside of the system, and making a precise compensation. For obtaining the information of control quantity and disturbance quantity, the further transformation equation about PMSM is as follows:
Where, represents the unobservable disturbance of the system.
According to the PMSM mathematical model, ADRC designed principle and vector control frame, a PMSM speed control system based on an improved ADRC is acquired, the control block diagram of PMSM speed control system based on ADRC is shown in Fig. 3.
According to Fig. 3, the ADRC variable related to control system quantity is realized, and PMSM speed control system will have better performance in robustness, the specific forms of variables in TD, ESO and NLSEF expressions are gained.
As to TD, the real meaning of the reference signal in controller refers to expected velocity ; The real meaning of output signal within control system symbolizes the tracking signal related to expected velocity based on TD. As to ESO, realistic importance for the output signal in control system means real speed ; the practical significance of the output signal in the control system is the actual speed tracking signal; The practical significance of the output signal in the control system is the estimation of the disturbance signal. As to NLSEF, realistic significance in input signal within control system can be evidenced by the error of 、、 and .
- What is the motivation of using of memetic algorithm.
Response:
- Thanks for your constructive comment to improve our manuscript!
- In the revised version, we have added the motivation of using of memetic algorithm as follows.
In fact, Memetic algorithm proposes a framework, which can be equal to a collaboration model between global population evolution and local individual learning. The model of Memetic algorithm is similar to genetic algorithm, however, under the premise of appropriate framework design, its global optimization performance far exceed genetic algorithm, for several specific optimization problems, the optimization precision could be improved to a considerable extent and several even orders of magnitudes [23]. For the motivation of improving the integration parameters optimization effect effectively, memetic algorithm is used in this paper, and its improvement strategies study is also be heeded.
- There is no evidence to use to show the superiority of the suggested algorithm; add some comparisons and analysis with new methods.
Response:
- Thanks for your constructive comment to improve our manuscript!
- In the revised version, the evidence of comparisons and analysis for comparing performance of the proposed control strategy (modified ADRC based on IMA) with other methods (fuzzy PI, traditional ADRC based on genetic algorithm and improved ADRC with effective strategies).
Finally, the experiment results of PMSM speed control practical cases show that the modified ADRC based on IMA has an apparent better optimization effect than that of fuzzy PI, traditional ADRC based on genetic algorithm and an improved ADRC based on improved moth-flame optimization.
- How the method can be improved using fuzzy controllers such as: Modeling renewable energy systems by a self-evolving nonlinear consequent part recurrent type-2 fuzzy system for power prediction.
Response:
- Thanks for your constructive comment to improve our manuscript!
- Obviously, combining with fuzzy controllers is profit for improving the control performance of PMSM, such as etc.: Modeling renewable energy systems by a self-evolving nonlinear consequent part recurrent type-2 fuzzy system for power prediction, Mirror milling trajectory planning for large thin-walled parts based on Fuzzy-ADRC controlled force pre-supporting. These contributions should be shown in the introduction. Furthermore, in the part of suggestions as future directions, the improvement strategy about combining with fuzzy controllers should also be expounded.
- In the revised version, the contribution introduction and future directions suggestions about combining with fuzzy controllers has been provided as follows.
A novel mirror milling trajectory planning method based on fuzzy-ADRC controlled force pre-supporting was proposed in [12]. A novel Nonlinear Consequent Part Recurrent Type-2 Fuzzy System (NCPRT2FS) was presented for the modeling of renewable energy systems in [13].
There are several future directions suggestions for ADRC of PMSM: Ⅰ draw lessons from the existing advanced control methods, such as robust predictive control, fuzzy predictive control or adaptive model predictive control, to further improve its control performance; Ⅱ designing a more suitable novel nonlinear optimal control function, so as to improve its control accuracy; Ⅲ combine the improved ADRC with fuzzy controller, to further improve its control performance Ⅳ construct a verification environment using the complex experiment so as to further improve the verification precision.

Reviewer 3 Report
In this paper, a new nonlinear function is constructed to realize the nondifferentiability and non-smooth condition of the nonlinear function at the piecewise point in the traditional ADRC. An improved meme algorithm based on an adaptive nonlinear decreasing convergence factor strategy is proposed to improve the control accuracy of permanent magnet synchronous motors. However, the article still needs to be revised, as follows:
1. Some formula symbols, such as Rs in (1), are not explained clearly. In addition, dq in symbol interpretation is not reflected in the formula, etc.
2. The author gives (1)–(4) four simple formulas in the 2.1 model building section. What is the final model like? What are the inputs and outputs in the model? It is not clearly stated in the text.
3. The controller design in 2.2 is more like designing a general controller and does not need the model established in 2.1. In this paper, the reader feels that the model building and controller design are two independent parts, and there is no effective connection.
4. How is the proposed method implemented? It is suggested to add the flow chart for control algorithm implementation.
5. In the third part, what is the purpose of the three existing optimization algorithms? Is the author writing a summary?
6. Figures 6–9: Is this the simulated image? What are the parameters of the simulation model?
7. In the control algorithm part, the author did not consider the problem of input and output constraints, so how can we ensure that the input and output of the system are bound?
8. In the analysis of Figures 6–9, the author only briefly described it, without further analysis. What is the reason for adding corresponding advantages?
9. In the introduction, the author only analyzes the traditional control methods, while some advanced control methods have not been analyzed. For example, the robust predictive control mentioned in doi: 10.1109/TFUZZ.2019.2959539 and doi: 10.1002/rnc.5712 also has strong processing capacity for uncertainties and disturbances.
Author Response
We sincerely thank you for the comments that helped a lot to improve the quality of our submission.
Overall evaluation: In this paper, a new nonlinear function is constructed to realize the nondifferentiability and non-smooth condition of the nonlinear function at the piecewise point in the traditional ADRC. An improved meme algorithm based on an adaptive nonlinear decreasing convergence factor strategy is proposed to improve the control accuracy of permanent magnet synchronous motors. However, the article still needs to be revised, as follows.
Response:
- Thanks for your overall evaluation!
- Some formula symbols, such as in (1), are not explained clearly. In addition, in symbol interpretation is not reflected in the formula, etc.
Response:
- Thanks for your constructive comment to mprove our manuscript!
- In the revised version, several formula symbols have been revised as follows.
The surface mounted structure is widely used in PMSM, due to approximate sine wave distribution for air gap flux density waveform about the motor can be obtained, and the inductance components of the direct axis (d axis) and the quadrature axis (q axis) of the motor can be equal, thus ensuring that the motor has a good operation performance.
is the motor resistance.
and are the direct and quadrature axial inductance.
The specific calculation formula of traditional optimal control function is as follows:
where is the error between the follow value of the TD to the reference signal and the estimated value of the system output signal by the ESO; is the differential of ; represents the output of NLSEF; represents the control rate; is the compensation coefficient to eliminate the interference signal.
- The author gives (1)-(4) four simple formulas in the 2.1 model building section. What is the final model like? What are the inputs and outputs in the model? It is not clearly stated in the text.
Response:
- Thanks for your constructive comment to mprove our manuscript!
- In the revised version, we have added the transformation equation about PMSM, inputs and outputs description as follows.
The PMSM speed control system based on ADRC is composed of an ADRC, an inverter circuit and a PMSM. The ADRC controls the speed and torque of the PMSM. The specific meaning of the input and output signals of each module in the PMSM speed control system needs to be determined according to the controlled object. The transformation equation about PMSM of equations (2) to (4) is as follows:
The greatest advantage of ADRC is its capability in estimating disturbances inside and outside of the system, and making a precise compensation. For obtaining the information of control quantity and disturbance quantity, the further transformation equation about PMSM is as follows:
Where, represents the unobservable disturbance of the system.
According to the PMSM mathematical model, ADRC designed principle and vector control frame, a PMSM speed control system based on an improved ADRC is acquired, the control block diagram of PMSM speed control system based on ADRC is shown in Fig. 3.
According to Fig. 3, the ADRC variable related to control system quantity is realized, and PMSM speed control system will have better performance in robustness, the specific forms of variables in TD, ESO and NLSEF expressions are gained.
As to TD, the real meaning of the reference signal in controller refers to expected velocity ; The real meaning of output signal within control system symbolizes the tracking signal related to expected velocity based on TD. As to ESO, realistic importance for the output signal in control system means real speed ; the practical significance of the output signal in the control system is the actual speed tracking signal; The practical significance of the output signal in the control system is the estimation of the disturbance signal. As to NLSEF, realistic significance in input signal within control system can be evidenced by the error of 、、 and .
- The controller design in 2.2 is more like designing a general controller and does not need the model established in 2.1. In this paper, the reader feels that the model building and controller design are two independent parts, and there is no effective connection.
Response:
- Thanks for your constructive comment to mprove our manuscript!
- In fact, the PMSM speed control system based on ADRC is composed of ADRC and decouple for vector control, and the specific implementation details demands knowledge about section 2.1 (model building) and section 2.2 (controller design)
- In the revised version, more implementation details about the PMSM speed control system based on ADRC have been provided as follows.
The PMSM speed control system based on ADRC is composed of an ADRC, an inverter circuit and a PMSM. The ADRC controls the speed and torque of the PMSM. The specific meaning of the input and output signals of each module in the PMSM speed control system needs to be determined according to the controlled object. The transformation equation about PMSM of equations (2) to (4) is as follows:
The greatest advantage of ADRC is its capability in estimating disturbances inside and outside of the system, and making a precise compensation. For obtaining the information of control quantity and disturbance quantity, the further transformation equation about PMSM is as follows:
Where, represents the unobservable disturbance of the system.
According to the PMSM mathematical model, ADRC designed principle and vector control frame, a PMSM speed control system based on an improved ADRC is acquired, the control block diagram of PMSM speed control system based on ADRC is shown in Fig. 3.
According to Fig. 3, the ADRC variable related to control system quantity is realized, and PMSM speed control system will have better performance in robustness, the specific forms of variables in TD, ESO and NLSEF expressions are gained.
As to TD, the real meaning of the reference signal in controller refers to expected velocity ; The real meaning of output signal within control system symbolizes the tracking signal related to expected velocity based on TD. As to ESO, realistic importance for the output signal in control system means real speed ; the practical significance of the output signal in the control system is the actual speed tracking signal; The practical significance of the output signal in the control system is the estimation of the disturbance signal. As to NLSEF, realistic significance in input signal within control system can be evidenced by the error of 、、 and .
- How is the proposed method implemented? It is suggested to add the flow chart for control algorithm implementation.
Response:
- Thanks for your constructive comment to mprove our manuscript!
- In the revised version, we have added the flow chart for proposed control algorithm as follows.
The design core of the ADRC based on improved memetic algorithm (IMA) is the integration parameters optimization mechanism for ADRC. Aiming at improving the integration parameters optimization effect, an adaptive nonlinear decreasing strategy for convergence factor, gaussian mutation mechanism, improved learning mechanism and an elite set maintenance mechanism based on fusion distance are integrated into the moth flame algorithm.
The flowchart for the proposed ADRC based on improved memetic algorithm is shown as follows.
- In the third part, what is the purpose of the three existing optimization algorithms? Is the author writing a summary?
Response:
- Thanks for your constructive comment to mprove our manuscript!
- In the revised version, the summary of purpose for the three existing optimization algorithms as comparison objects has been provided as follows.
In order to verify the performance of NewADRC-IMA proposed in this paper, PMSM velocity control experiments are implemented. Fuzzy PI and traditional ADRC based on genetic algorithm are two popular traditional control algorithms, they are widely used in PMSM velocity control due to the characters of being stable in control and easy for realizing them, however, their tracking control accuracy is relatively low. In this paper, traditional ADRC based on genetic algorithm is abbreviated as ADRC-GA. An improved ADRC with effective strategies was proposed to solve the defect that the nonlinear function of traditional ADRC is not differentiable at the piecewise point, and an improved moth-flame optimization (MFO) was proposed to obtain its key parameters, it can effectively improve the tracking control performance. The above specific modified ADRC is abbreviated as NADRC-IMFO.
The performance of proposed control algorithm NewADRC-IMA is necessary to be verified, and three control algorithms (fuzzy PI, ADRC-GA and NADRC-IMFO) are used for comparison.
- Figures 6–9: Is this the simulated image? What are the parameters of the simulation model?
Response:
- Thanks for your constructive comment to mprove our manuscript!
- Obviously, section 5.3 presents the experiment result and analysis.
- In the revised version, the parameters about ADRC based on IMA has been provided as follows.
In this paper, as far as modified NewADRC redesigned in this paper, the optimization integration of ADRC key parameters has been obtained by IMA, the specific ADRC is abbreviated as NewADRC-IMA. Detailed parameters of improved IMA are shown below: the population size is set as 40, the maximum number of iteration times is set as 80, the probability of the surrounding preys is set as 0.6, the adaptive nonlinear decreasing optimization factor for convergence factor is set as 1.75, the probability of mutation selection behavior is set as 0.15, the initial temperature is 200℃, the cooling coefficient is 0.75, and the termination temperature is 50℃, the elit size is set as 25. Detailed results about the optimization integration of ADRC key parameters by using IMA are shown below: obtained optimization integrated parameters、、、、 and .
- In the control algorithm part, the author did not consider the problem of input and output constraints, so how can we ensure that the input and output of the system are bound?
Response:
- Thanks for your constructive comment to mprove our manuscript!
- In the revised version, the section 2.4 about evaluation model (performance evaluation indexes and boundary constraints) of PMSM speed control has been rewritten as follows.
The integral of time multiplied by the absolute value of error (ITAE) is one of the significant evaluation index for ADRC control performance. The specific calculation formula of ITAE is as follows:
During PMSM speed tracking control process, the time about the long-term control range with the acceptable maximum absolute value of stable control speed and torque error ( and ) is called as stable time , the above referred error should be less than acceptable maximum stable control speed and torque error ( and ); the time between the begin time point of power-up and the corresponding begin time point of stable control range is called as adjusting time , adjusting time must not exceed maximum acceptable adjusting time ; the maximum absolute value of speed and torque error ( and ) should be less than maximum acceptable control speed and torque error ( and ). The specific evaluation model of PMSM speed control is as follows:
where , , , , and are 6 performance evaluation indexes for optimization.
Based on these above performance evaluation indexes and boundary constraints, the diagram about the above parameters adjusting method using intelligent optimization algorithm (IOA) is shown in Fig. 3.
- In the analysis of Figures 6–9, the author only briefly described it, without further analysis. What is the reason for adding corresponding advantages?
Response:
- Thanks for your constructive comment to mprove our manuscript!
- In the revised version, the section 4.1 about design of a novel differentiable and smooth nonlinear optimal control function has been rewritten, the section 4.1 about design of ADRC based on improved memetic algorithm has been added, and the reason analysis and summary for adding corresponding advantages.
Clearly, the improved strategies introduced in this paper are effective. The nonlinear optimal control function constructed in this paper are differentiable and smooth at the piecewise point for modifying ADRC, and IMA proposed in this paper has powerful global optimization capability, so the modified ADRC based on IMA is improved obviously.
- In the introduction, the author only analyzes the traditional control methods, while some advanced control methods have not been analyzed. For example, the robust predictive control mentioned in doi: 10.1109/TFUZZ.2019.2959539 and doi: 10.1002/rnc.5712 also has strong processing capacity for uncertainties and disturbances.
Response:
- Thanks for your constructive comment to mprove our manuscript!
- Obviously, some advanced control methods should be analyzed, such as etc.: doi: 10.1109/TFUZZ.2019.2959539 (Robust Fuzzy Predictive Control for Discrete-Time Systems With Interval Time-Varying Delays and Unknown Disturbances ), doi: 10.1002/rnc.5712 (Robust adaptive model predictive control for guaranteed fast and accurate stabilization in the presence of model errors). These contributions should be shown in the introduction. Furthermore, in the part of suggestions as future directions, the improvement strategy about some advanced control methods and future directions suggestions about combining with fuzzy controllers should also be expounded.
- In the revised version, the contribution introduction and future directions suggestions about robust predictive control mentioned has been provided as follows.
For systems with uncertainties, time-varying delays, unknown disturbances, as well as strong nonlinearity, a robust fuzzy predictive control (RFPC) based on Takagi-Sugeno (T-S) fuzzy model was proposed in [14]. A robust adaptive model predictive control (RMPC) with an underlying discrete-time adaptive controller was proposed in [15].
There are several future directions suggestions for ADRC of PMSM: Ⅰ draw lessons from the existing advanced control methods, such as robust predictive control, fuzzy predictive control or adaptive model predictive control, to further improve its control performance; Ⅱ designing a more suitable novel nonlinear optimal control function, so as to improve its control accuracy; Ⅲ combine the improved ADRC with fuzzy controller, to further improve its control performance Ⅳ construct a verification environment using the complex experiment so as to further improve the verification precision.

Round 2
Reviewer 1 Report
Accept in present form
Reviewer 2 Report
The paper can be accepted in my opinion
Reviewer 3 Report
The format of the reference should be checked carefully, such as the the name of the author.